# The urinary microbiome in association with diabetes and diabetic kidney disease: A systematic review

Tiscar Graells[1]*, Yi-Ting Lin[1,2], Shafqat Ahmad[3,4,5], Tove Fall[3], Johan Ärnlöv[1,6,7]

1 Department of Neurobiology, Care Sciences and Society, Division of Family Medicine and Primary Care, Karolinska Institute, Huddinge, Stockholm, Sweden, 2 Department of Family Medicine, Kaohsiung Medical University Hospital, Kaohsiung Medical University, Kaohsiung, Taiwan, 3 Molecular Epidemiology, Department of Medical Sciences, Uppsala University, Uppsala, Sweden, 4 Preventive Medicine Division, Harvard Medical School, Brigham and Women's Hospital, Boston, Massachusetts, United States of America, 5 Natural Sciences, Technology and Environmental Studies, Södertörn University, Huddinge, Sweden, 6 Center for Clinical Research Dalarna, Uppsala University, Falun, Sweden, 7 School of Health and Welfare, Dalarna University, Falun, Sweden

* tiscar.graells.fernandez@ki.se

**Data Availability Statement:** Data presented and gathered in this systematic review are fully shown in this manuscript or in the supplementary information.

## Abstract

### Background

The urinary microbiome, or urobiome, is a novel area of research that has been gaining attention recently, as urine was thought to be sterile for years. There is limited information about the composition of the urobiome in health and disease. The urobiome may be affected by several factors and diseases such as diabetes, a disease that often leads to kidney damage. Thus, we need to understand the role of the urobiome to assess and monitor kidney disease related to diabetes over time.

### Methods

We conducted a systematic review to summarize knowledge about the urobiome in association with diabetes mellitus and diabetic kidney disease. The search was conducted in several electronic databases until November 2024.

### Results

Eighteen studies were selected including cross-sectional case-control studies, cross-sectional surveys and one prospective longitudinal study. In total, the urobiome of 1,571 people was sequenced, of which 662 people had diabetes, and of these 36 had confirmed diabetic kidney disease; 609 were healthy individuals, 179 had prediabetes or were at risk of type 2 diabetes mellitus and 121 did not have diabetes but had other comorbidities. Eight studies analysed data from females, one was focused on male data, and the other nine had mixed female-male data. Most of the studies had a small sample size, used voided midstream urine, and used 16S rRNA sequencing.

**Funding:** We acknowledge funding from the Swedish research council (Vetenskapsrådet grant number 2019-01015 (J.Ä.), 2019-01471 (T.F.) and 2020-0243 (J.Ä.)), from the Swedish heart lung foundation (Hjärt-Lungfonden grant number 2021-0357 (J.Ä.), 2023-0687 (T.F.) and 2024-0486 (J.Ä.)), and from Center of clinical research (CKF) in Region Dalarna, Falun, Sweden. Dr Ahmad was supported by research grants from FORMAS–Early Career Grant (no. 2020-00989), Swedish Research Council-Early Career Grant (no. 2022-01460), EFSD/Novo Nordisk and EpiHealth. The funders had no role in study design, data collection and analysis, decision to publish, or preparation of the manuscript.

**Competing interests:** J.Ä. has served on the advisory boards for Astella, AstraZeneca, and Boehringer Ingelheim and has received lecturing fees from AstraZeneca and Novartis, all unrelated to the present work. This does not alter our adherence to PLOS ONE policies on sharing data and materials. The remaining authors declare no competing interests. There are no patents, products in development or marketed products associated with this research to declare.

## Conclusion

This systematic review summarizes trends seen throughout published data available to have a first baseline knowledge of the urinary microbiome, and its microbiota, in association with diabetes including the decreased richness and α-diversity in urinary microbiota in individuals with diabetes compared to healthy controls and the decreased α-diversity with the evolution of kidney disease independently of the cause.

## 1. Introduction

Microbial communities living within us comprise viruses, bacteria, archaea, and small eukaryotes like fungi and protists [1]. These communities are symbiotic microbial networks that can perform vital functions in the host and whose metabolites, or the lack of them, have influence in human health and diseases [2–4]. The intricate structure of these microbial ecosystems within their ecological niche, their taxonomy and functional composition, are known as the microbiome [5]. Similarly to our fingerprints, each microbiome in the gut, vagina, skin or urinary tract are highly unique and they are influenced by the environment, host genetics, lifestyle and there may be other factors influencing the microbiome for which we currently lack understanding [4]. Dysbiosis is a state where the microbiome is disrupted or altered, or both, impeding normal functionality or causing low-grade inflammation that may play a role in the onset and/or development of several diseases [5]. The human gut microbiome is seen as a "hidden" metabolic organ for human wellbeing affecting immunity, neural, endocrinal or metabolic pathways [4, 6]. Gut dysbiosis has been related to several diseases but if it is cause or effect of these diseases is a question that remains unclear [4, 7, 8]. While our understanding of the gut microbiome is evolving but limited, the knowledge about other microbiomes is even more scarce. This is particularly true for the urinary tract microbiome, or urobiome, which was long believed to be sterile until a decade ago [9, 10]. The urobiome is characterized by a low biomass relative to the gut microbiome, and its composition remains largely unexplored [9, 11]. Evidence suggests that the urobiome differs between males and females: associated with lactobacilli in females, whereas in males is linked to *Corynebacterium* or *Staphylococcus* [9, 11, 12].

Diabetes, a disease characterized by elevated levels of glucose in the blood, is a leading cause of cardiovascular disease, blindness or kidney damage [13]. Diabetes, especially type 2 diabetes mellitus (T2DM), has become a global health emergency whose prevalence have been rising for decades affecting 537 million people globally in 2021, including 6.7 million related deaths [13, 14]. The incidence of diagnosed diabetes in adults is stabilising in some high-income countries, but incidence in children is still increasing [15–18]. Impaired glycaemia contributes to enhanced risk of infections, and it may explain why diabetic individuals have higher risk of urinary infections, pyelonephritis and urosepsis [19]. Not only individuals with diabetes have higher glucose, albumin and other proteins in urine which may affect the urobiome composition, enhance bacterial growth of some species and may influence the diversity of the urobiome; but also they have high glucose in urine which compromises the immune response and uroepithelial integrity [19]. However, we have limited information about the composition of the urobiome related to diabetes. In addition, damage in the kidney blood vessels, because of diabetes or other causes, may decline the estimated glomerular filtration rate (eGFR) which impacts urinary functions as these organs are connected through the ureters to the urinary bladder [20]. Development of diabetic nephropathy may further turn in chronic kidney disease (CKD) or end stage renal disease (ESRD), the former associated with diabetes but also other comorbidities while the latter is mostly caused by diabetes [20].

As there is a clear knowledge gap regarding the role of the urobiome in diabetes and diabetic kidney disease, the aim of this systematic review is to gather available data and synthetise the main findings about the urobiome, particularly, the urinary microbiota in association with diabetes and kidney conditions derived from diabetes such as CKD or ESRD.

## 2. Materials and methods

### 2.1 Search strategy

The comprehensive literature search was conducted according to the Preferred Reporting Items for Systematic Reviews and Meta-analyses (PRISMA) guidelines [21]. It was registered under accession number of 565545 in the International Prospective Register of Systematic Reviews (PROSPERO) hosted by the National Institute for Health and Care Research of the University of York (United Kingdom). A comprehensive literature search on different databases was conducted from 1$^{st}$ January 2000 until 28$^{th}$ November 2024. The search included the following terms [("urobiome" OR "urinary microbiome" OR "urinary microbiota" OR "urinary tract microb*" OR "urine microb*" OR "urogenital microbiome" OR "urogenital microbiota") AND ("kidney disease*" OR "chronic kidney disease" OR "diabetes" OR "albuminuria" OR "diabetic nephropathy" OR "end stage renal disease*" OR "renal disease*")]. We used the 'TITLE-ABS-KEY' search in SCOPUS and Cochrane Central Register of Controlled Trials databases, the 'Keyword' in advanced search in MEDLINE-Ovid and EMBASE databases, while 'ALL FIELDS' were used in Web of Science and PUBMED databases (S1 Table in S1 File). This search wanted to identify relevant clinical urinary microbiome studies associated with diabetes and/or kidney impairment published from 1$^{st}$ January 2000 until 28$^{th}$ November 2024.

### 2.2 Study selection

We included studies that evaluated the urobiome sequencing any type of urine samples (urinary bladder, urinary kidney or urogenital samples [22]) in relation with diabetes and conditions that affect kidney functionality, and which can lead to CKD or ESRD. Studies were excluded if they were review articles, editorials, comments, systematic reviews, books, conference abstracts or any other type of article without full data available. The inclusion criteria were: (i) analyses studying the bacterial species of urine; (ii) analyses of individuals with diabetes and/or albuminuria and/or diabetic CKD and/or diabetic ESRD; (iii) human clinical samples; and (iv) perform DNA sequencing through 16S ribosomal DNA gene or shotgun next generation sequencing.

### 2.3 Data extraction, assessment of quality and risk of bias

Throughout this synthesis, we used the PRISMA and PROSPERO frameworks and we incorporated the associated guidance at each stage. Following the removal of duplicates, the first step was the screening of studies by article type. After removal of non-suitable studies, the second step included a screening of studies by title and/or abstract for relevance. Those that seemed relevant were the preliminary selected publications and they were finally screened based on full-text information. Studies lacking full-text availability or with missing or unclear information were excluded from the evaluation (Fig 1). Two independent authors (T.G and Y.T.L) screened studies by title and abstract and, if discrepancies arose, they were discussed until they reached a common agreement. Preliminary selected publications were fully screened by T.G., reviewed by Y.T.L, and if they fulfilled the inclusion criteria, they were selected for this systematic review. Studies quality and risk of bias were assessed on participant selection,

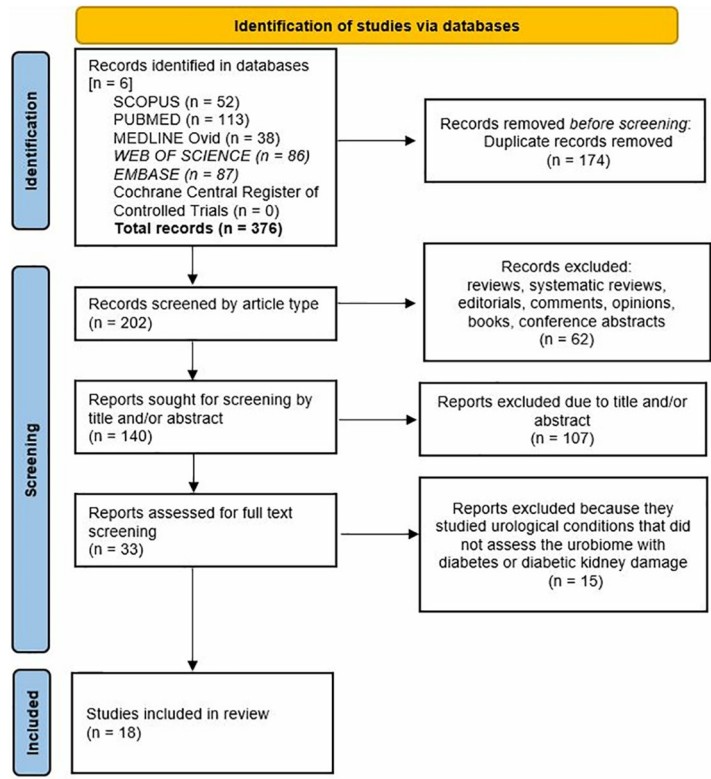

**Fig 1. Flow diagram of selected studies.** Diagram showing how studies in this systematic review about the urinary microbiota and diabetes were selected.

exclusion criteria, data measurements and analysis [23, 24]. A determination of "low risk" or "high risk" bias according to the Cochrane and the National Heart, Lung and Blood of the United States (NHLBI) guidelines was assessed by T.G. and independently checked by Y.T.L and J.Ä. [23, 25]. Any disagreement was discussed and resolved between these authors.

## 3. Results

### 3.1 Selected studies

In total, eighteen studies were included including cross-sectional case-control studies, cross-sectional surveys and one prospective longitudinal study (Table 1 and S2 Table in S1 File). These studies altogether sequenced the urinary microbiome of 1,571 people, some participants were part of the same cohort studied and were included multiple times [26–29]. Of these individuals 662 had diabetes, 609 were healthy individuals, 179 had prediabetes or were at risk of diabetes, and 121 did not have diabetes but had other comorbidities. Among the 662 diabetic individuals, 36 individuals had diabetic kidney disease [30–32] while one studied CKD patients, most of them with diabetes, but did not mention the primary cause of CKD [33]. Sixteen of the studies used voided midstream urine samples, or a modification of this collection technique, to avoid as much as possible skin or genitalia microorganisms (S2 Table in S1 File). The remaining two studies analysed urinary bladder microbiota (catheterised urine) and, among these two, one was focused only on males (Table 1). Eight studies included only female samples (four of these used the same cohort data); and nine studies had mixed female-male data (Table 1). All studies except one used 16S sequencing which included twelve studies with

**Table 1.** Characteristics and results of selected studies.

| Article | Country | Total adult | Healthy vs diabetic adults | Age: mean ± SD | Context | Indexes results in diabetic individuals | Main findings and/or differences in relative abundance of diabetic individuals in urine | Other significant details or relevant information |
|---|---|---|---|---|---|---|---|---|
| Case-control studies with controls and diabetic individuals | | | | | | | | |
| [34] | China | 30 | 0 vs 15 | Kidney stones and T2DM: 56 ± 11 Kidney stones alone: 55 ± 12 | Urinary microbiota in the renal pelvis of patients with kidney stones plus T2DM and those with kidney stones alone | No differences in richness (Chao index) ↑ α—diversity (Shannon index) β-diversity: the group with kidney stones alone and with kidney stones and T2DM clustered and could be differentiated | ↑Sphingomonas ↑Propionibacterium ↑Corynebacterium ↑Cellulosimicrobium ↑Methylophilus ↑Lactobacillus ↑Enhydrobacter ↑Chryseobacterium ↑Haemophilus ↑Allobaculum | The relative abundance of Enhydrobacter, Chryseobacterium, and Allobaculum genera exhibited correlations with fasting blood glucose and HbA1c values. |
| [42] | Italy | 75 | 25 vs 50 | H males: 62 ± 7 H females: 57 ± 6 T2DM males: 65 ± 9 T2DM females: 64 ± 10 | | No differences in richness or α—diversity (observed species, Chao, and Shannon indexes) | NS [Statistical differences in prevalence (males and females): Bifidobacterium breve, Corynebacterium pyroviciproducens, Lactobacillus gasseri, and Streptococcus agalactiae were more prevalent in healthy controls. Campylobacter ureolyticus, Corynebacterium coyleae, Corynebacterium glucuronolyticum, Enterococcus faecalis, Escherichia coli, Lactobacillus iners, Peptoniphilus grossensis, and Veillonella atypica were more prevalent in T2DM individuals] | Total bacterial load and the abundance of total Bacillota were found to be elevated in patients with diabetes compared with the healthy control group, this was also true for females but not males. Actinotignum schaalii, Bifidobacterium scardovii, Facklamia hominis, Negativicoccus succinicivorans, and Peptoniphilus lacrimalis were less prevalent in T2DM males. Aerococcus christensenii, Anaerococcus hydrogenalis, Brevibacterium ravenspurgense, Corynebacterium aurimucosum, Gardnerella vaginalis, Mobiluncus curtisii, Prevotella buccalis, Prevotella colorans, and Veillonella montpellierensis were more prevalent in T2DM females whereas Facklamia ignava and Winkia neuii were less prevalent in T2DM females. |
| [31] | China | 34 | 15 vs 19 | H: 44 ± 8 T2DM: 60 ± 9 | T2DM with diabetic kidney disease | ↓ α—diversity in T2DM (Shannon and Simpson indexes) | ↓Bacillota ↓Bacteroidota ↑Pseudomonadota ↑Acidobacteriota | Clinical characteristics between H and T2DM differed significantly in age; BMI; SBP; urea; uric acid; creatinine; natriuretic peptide; HbA1c (all ↓in H); and HDL and albumin (both ↑ in H). HbA1c was positively correlated with Acidobacteria and eGFR was positively correlated with Bacillota, Clostridia, Clostridiales, Lactobacillaceae and Lactobacillus. |
| [43] ᶠ | South Korea | 691 | 368 vs 164 | 59 ± 6 (average of all groups and periods) | 4-year longitudinal study of sequencing extracellular microbial vesicles. Also, 164 prediabetics and 35 people transitioned | ↓ α—diversity (Shannon index) in all groups (possible effect of aging) | ↓ unclassified Lachnospiraceae GU174097_g | Low abundance of GU174097_g was a risk factor for T2DM development, and it was associated with T2DM progression. GU174097_g decreased ketone bodies levels; and thus reduced HbA1c levels. |

(Continued)

**Table 1.** (Continued)

| Article | Country | Total adult | Healthy vs diabetic adults | Age: mean ± SD | Context | Indexes results in diabetic individuals | Main findings and/or differences in relative abundance of diabetic individuals in urine | Other significant details or relevant information |
|---|---|---|---|---|---|---|---|---|
| [37] ♂ Ω | Czech Republic | 58 ♂ | 0 vs 14 | 65 ± 14 (all) | Catheterized urine samples from men collected under anesthesia from the bladder prior to urological surgery. 14 individuals had diabetes (36 did not have diabetes and for 8 it was unknown) | ↓richness (OTUs, ACE and iChao2 indexes) ↓ α—diversity (Shannon and Simpson indexes) β—diversity was no different | No statistical differences of specific taxa between T2DM and non-T2DM individuals | Lower richness was also associated with: (1) ≥75 years; (2) High cholesterol and/or hyperlipidemia; (3) antibiotic prophylaxis. Smokers had higher α—diversity. No significant differences in α—diversity for hypertension or CKD. β–diversity was different between individuals with (1) CKD vs no CKD; (2) severe vs mild urinary symptoms; and (3) antibiotic prophylaxis vs without |
| [40] ¥ | Turkey | 60 | 15 vs 15 | H: 42 ± 11 Obese: 42 ± 9 Pre-DM: 50 ± 9 T2DM: 54 ± 8 | T2DM, prediabetic (n = 15), obese (n = 15) and H groups | NS | ↓ *Bifidobacterium* (same was observed for obese and prediabetics) | Clinical features statistically different: FBG between all groups; HbA1c between all groups except for H and obese individuals; BMI between all groups except for prediabetics and both obese and T2DM. Age was significantly lower in both H and obese vs T2DM individuals. Triglycerides and HDL were significantly lower in H vs both prediabetics and T2DM individuals |
| [36] ♀ | United States | 136 ♀ | 49 vs 87 | H: 51 ±11 T2DM: 51 ± 11 | Females with T2DM | β–diversity: Both H and T2DM clustered in 4–5 urotypes Urotypes in T2DM: *Lactobacillus* and *Enterobacteriaceae* vs H females: *Gardnerella* and mixed | ↑ *Lactobacillus* ↓ *Corynebacterium* ↓ *Staphylococcus* [T2DM differences in presence/absence: *Lactobacillus* was more present while *Corynebacterium, Anaerococcus, Finegoldia* and *Peptoniphilus* were less present] | H and T2DM groups differed significantly in FBG and HbA1c (both ↓in H). HbA1c was positively correlated with *Lactobacillus* and negatively associated with *Prevotella* and *Corynebacterium* in all cohort; and negatively associated with *Prevotella* also in the T2DM group. Prevalence of *Lactobacillus* increased with weight; and most of participants were obese. |

*(Continued)*

**Table 1.** (Continued)

| Article | Country | Total adult | Healthy vs diabetic adults | Age: mean ± SD | Context | Indexes results in diabetic individuals | Main findings and/or differences in relative abundance of diabetic individuals in urine | Other significant details or relevant information |
|---|---|---|---|---|---|---|---|---|
| [35] ♀ | China | 58 ♀ | 26 vs 32 | H: 58 ± 9 T2DM: 57 ± 8 | Females H and T2DM. T2DM microbiota were also assessed for lower urinary symptoms and levels of HbA1c | Richness and α–diversity were no different (OTUs, observed species, Chao, ACE, Shannon and Simpson indexes) β—diversity was different between H and T2DM | ↑ Escherichia-Shigella ↑Klebsiella ↑Aerococcus ↑Delftia ↑Enterococcus ↑Stenotrophomonas ↑Micrococcus ↑Deinococcus ↓Gallicola ↓Arcobacter ↓Arcanobacterium ↓Kocuria ↓Murdochiella ↓Solitalea ↓Peptoniphilus | FBG was ↑ in T2DM compared to H group. HbA1c was ↑ in T2DM with urinary symptoms vs T2DM without. The diabetic group with urinary symptoms (compared to the group without) had lower richness, different β–diversity; and: ↑Escherichia-Shigella ↑Campylobacter ↑Megasphera ↓Prevotella ↓Dialister ↓Anaerococcus ↓Fusobacterium ↓Prevotella_6, ↓Mycoplasma ↓Peptoniphilus ↓Porphyromonas ↓Fodinicola ↓Brevundimonas The diabetic group with high HbA1c compared to the diabetic low HbA1c group had lower α–diversity, different β–diversity; and: ↑Escherichia-Shigella ↑Lactobacillus ↓Prevotella ↓Campylobacter ↓Dialister ↓Anaerococcus ↓Peptoniphilus ↓Porphyromonas ↓Fodinicola |
| [39] ♀ | China | 100 ♀ | 50 vs 50 | Elderly: 72 ± 7 Non-old: 50 ± 8 | Elderly vs non-elderly cohorts. Half of the females of each cohort had T2DM | ↓ richness (OTU and Chao indexes) in elderly with T2DM No differences in α—diversity (Shannon and Simpson indexes) between the elderly with and without T2DM | Elderly with T2DM (compared to elderly without): ↓Nitrospirae ↓Odoribacter ↓Aeromonas ↓Clostridium ↓Enterobacter ↓Butyricimonas ↓Enterococcus ↑Erwinia ↑Fusobacterium ↓Klebsiella ↓Lachnobacterium ↓Stenotrophomonas ↓Phascolarctobacterium ↑Eggerthella ↑Parvimonas ↑Bdellovibrio ↑Hydrogenophaga ↑**Lactobacillus iners** | T2DM females treated with metformin. Lactobacillus had a decreased trend in T2DM patients with FBG > 10 mmol/l. Lactobacillus iners was not associated with FBG. Elderly related differences: (1) relative abundance of Bacillota (Firmicutes) increased with BMI; (2) relative abundance of Bifidobacteria was negatively correlated with age; (3) Lactobacillus decreased with age but was not associated with pH; (4) ↑ abundance of Sphingobium and Bosea; and (5) ↓ abundance of Sneathia; Geobacillus; Shuttleworthia; Bacillus; Gemella; Bdellovibrio; Hydrogenophaga; Proteus; Novosphingobium. |

(Continued)

**Table 1.** (Continued)

| Article | Country | Total adult | Healthy vs diabetic adults | Age: mean ± SD | Context | Indexes results in diabetic individuals | Main findings and/or differences in relative abundance of diabetic individuals in urine | Other significant details or relevant information |
|---|---|---|---|---|---|---|---|---|
| [26] ♀ | China | 140 ♀ | 70 vs 70 | Matched H and T2DM individuals. 70% > 56 years or older; only 8% younger than 45 years | Authors matched individual characteristics between the H and the T2DM groups | ↓ richness (number of OTUs, observed species, ACE and Chao1 indexes) ↓ α-diversity (Shannon index) | ↑ Actinobacteria ↑ Porphyromonas ↑ Flavobacteriales ↑ Collinsella ↓ Akkermansia muciniphila ↓ Bacteroidota ↓ Prevotellaceae ↓ Prevotella ↓Pseudomonadales ↓Peptoniphilus ↓ Citrobacter ↓Actinobacteriota ↓ Synergistota ↓Acinetobacter ↓Campylobacter ↓Campylobacterales ↓ Staphylococcus ↓ Anaerococcus ↓ Halomonas ↓Finegoldia ↓ Streptococcus ↓Veillonella ↓Clostridium ↓Corynebacterium | Actinobacteria increase was associated with higher BMI; increased FBG and urine glucose. Akkermansia muciniphila decrease was associated with FBG and urine glucose. Urine glucose, FBG, urine infections, hypertension and hyperlipidemia were statistically ↑ in T2DM compared to H group. Authors studied hypertension and hyperlipidemia in article by [28]; inflammatory markers in article by (Ling et al. 2017) and the effect on diet in article 7 [29]. |
| [32] Ω | United States | 29 | 8 vs 6 | H: 35 ESDR T1DM: 51 ESDR T2DM: 64 (all ESRD: 53) | Transplant ESRD patients. Kidney damage caused by 4 primary diagnoses: other, T1DM, T2DM, and hypertension. 15 individuals had non-diabetic ESRD. Trimethoprim -sulfamethoxazole was given prophylactically | ↓ α-diversity (Shannon and inverse Simpson indexes) ↓ richness (observed species) β-diversity: primary diagnoses had no significant differences in bacterial composition; but all had with the H | ↑ Enterococcus faecalis ↑ Enterococcus faecium ↑ Enterococcus sp. ↑ Escherichia coli ↑ Escherichia sp. ↓ Cutibacterium acnes ↓ Corynebacterium ↓Mobiluncus curtisii | ↑ Bacillota and ↓Actinobacteria because Enterococcus belongs to the former phylum while as Mobiluncus belongs to the latter. Both groups had similar folate metabolism (which is the target of trimethoprim -sulfamethoxazole) but had differences in specific enzymes. Dihydrofolate synthase/ folypolygluta-mate synthase was increased in transplant groups. |
| Studies with diabetic individuals only | | | | | | | | |
| [41] | Taiwan | 7 | 0 vs 7 | T2DM No-pyuria: 59 ± 6 T2DM Pyuria: 60 ± 7 | All 7 individuals with T2DM and SGLT2 therapy; 3 with pyuria in urine; 4 without | ↓ α—diversity in T2DM with pyuria compared with T2DM without pyuria (Shannon index) | ↑Escherichia-Shigella in individuals with pyuria | 7 T2DM individuals with negative standard urine cultures. eGFR stage II (60–89 mL/min) mildly impaired kidney function |
| [38] ♀ | China | 30 ♀ | 0 vs 30 | T2DM with diabetic peripheral neuropathy: 56 ± 8 T2DM only: 55 ± 7 | Among the 30 females with T2DM, 17 females also had diabetic peripheral neuropathy | ↓richness in DPN group (OTUs, observed species, ACE and iChao2 indexes) α–diversity was no different (Shannon and Simpson indexes) β–diversity was no different | Group with DPN: ↑ Mycoplasmataceae ↓Propionibacteriaceae ↑ Pseudobdellovibrioinaceae ↓Sphingobacteriaceae | T2DM females: 17 with DPN and 13 without. Both groups did not differ in clinical characteristics. |

(Continued)

**Table 1.** (Continued)

| Article | Country | Total adult | Healthy vs diabetic adults | Age: mean ± SD | Context | Indexes results in diabetic individuals | Main findings and/or differences in relative abundance of diabetic individuals in urine | Other significant details or relevant information |
|---|---|---|---|---|---|---|---|---|
| *♀ [28] | China | 70 ♀ | 0 vs 70 | T2DM: 56 ± 14 T2DM + hypertension: 70 ± 9 T2DM + hyperlipidemia: 54 ± 11 T2DM + both: 70 ± 10 | Only analysis of the T2DM individuals from article 13 that had different clinical features. The T2DM cohort was divided: 28 had T2DM; 24 had T2DM + hypertension; 7 had T2DM + hyperlipidaemia; 11 had T2DM and both hypertension + hyperlipidaemia | ↑ richness compared to the hyperlipidemia only group (number of reads and number of OTUs) ↓ richness compared to hypertension groups (number of reads and number of OTUs) No differences in α–diversity (Shannon and Simpson indexes) between the groups | Compared to T2DM + hypertension group, the group **with only** T2DM had: ↑ *Lactobacillus iners* ↑ *Acinetobacter rhizosphaerae* ↑ *Acinetobacter schindleri* ↑*Lactobacillus* ↑*Enterobacter* ↑*Klebsiella* ↑*Shuttleworthia* ↑*Sneathia* ↑*Parvimonas* ↑*Megasphera* ↑*Erwinia* ↓ *Kocuria palustris* ↓ *Aeromonas* ↓ *Roseburia* ↓ *Ruminococcus* Compared to T2DM + hyperlipidemia, the group **with only** T2DM had: ↓ *Faecalibacterium* ↓ *Collinsella* ↓ *Oscillospira* (diagnostic factor to differentiate both groups) Compared to T2DM + hypertension and hyperlipidemia, the group **with only** T2DM had: ↑*Acinetobacter rhizosphaerae* ↑ *Shuttleworthia* ↓*Gemella* ↓*Prevotella* Compared to T2DM + hypertension and hyperlipidemia, the group with T2DM + hyperlipidemia had: ↑*Acinetobacter rhizosphaerae* ↑*Faecalibacterium* Compared to T2DM + hypertension, the group with T2DM + hyperlipidemia had: ↑ *Klebsiella* ↑ *Novosphingobium* ↑ *Lactobacillus* ↑ *Enterobacter* The two groups with hypertension did not show differences | The T2DM cohort was divided: 28 T2DM; 24 T2DM + hypertension; 7 T2DM + hyperlipidemia; 11 T2DM + hypertension + hyperlipidemia. The duration of T2DM was significantly shorter in the two groups without hypertension. The four most abundant bacteria in all groups were Proteobacteria, Bacillota, Bacteroidota and Actinobacteria. The fifth varied from Fusobacteria (inT2DM and T2DM with both conditions), Synergistota (group with hypertension) and Acidobacteria (group with hyperlipidemia). Predominant genera: in T2DM *Lactobacillus, Prevotella,* and *Acinetobacter;* in T2DM + hyperlipidemia *Lactobacillus, Prevotella and Halomonas;* in T2DM + hypertension *Prevotella, Streptococcus,* and *Bacteroides;* and in T2DM + hyperlipidemia *Prevotella, Lactobacillus* and *Bacillus.* The groups with hypertension had presence of *Eggerthella lenta* but this species was absent in the group with only diabetes; while *Gardnerella* was absent in the hyperlipidemia group but it was present in the other groups. Several bacteria correlated with FBG, blood pressure and lipidic profiles. *Atopobium* was positively correlated with FBG in the T2DM group. *Allobaculum* and *Odoribacter* were negatively correlated with diastolic pressure in the hypertension and hypertension plus hyperlipidemia groups, respectively. *Rikenella* and *Dorea* were positively correlated with triglycerides in the hyperlipidemia group and with LDL in the hypertension plus hyperlipidemia group, respectively. |

(*Continued*)

Table 1. (Continued)

| Article | Country | Total adult | Healthy vs diabetic adults | Age: mean ± SD | Context | Indexes results in diabetic individuals | Main findings and/or differences in relative abundance of diabetic individuals in urine | Other significant details or relevant information |
|---|---|---|---|---|---|---|---|---|
| *♀ [27] | China | 70 ♀ | 0 vs 70 | T2DM with IL-8 in urine: 66 ± 14 T2DM without IL-8 in urine: 59 ± 11 | T2DM females whose levels of IL-8 were evaluated. IL-8 is a potential biomarker for diagnosing of urinary infections | α–diversity was no different between T2DM females with and without IL-8 in urine (Shannon and Simpson indexes) β-diversity: the group with Il-8 and without IL-8 in urine clustered together and could be differentiated | T2DM females without IL-8: ↓Bifidobacteriaceae ↓Shuttleworthia ↓Thermales ↓Streptococcus anginosus ↓Acinetobacter rhizosphaerae ↓Acinetobacter schindleri ↓Lactobacillus iners ↓Akkermansia muciniphila ↓Mobiluncus ↓Peptoniphilus ↓Corynebacterium ↓Gemella ↓Enterococcus ↓Geobacillus ↑Prevotella copri ↑Prevotella stercorea ↑Bacteroides uniformis ↑Coprococcus eutactus ↑Faecalibacterium ↑Megamonas ↑Comamonas ↑Pseudomonas ↑Phascolarctobacterium | Ruminococcus was positively associated with urinary IL-8. Urinary IL-8 was high in T2DM patients with ↑ Pseudomonas and ↑ Klebsiella. Clinical features between the T2DM with IL-8 and without IL-8 in urine differed in age, BMI and pH, white blood cells, leukocyte esterase, protein, glucose, and nitrites in urine (all characteristics ↑ in the group with IL-8) |
| *♂♀ [29] | China | 140 ♀ | 70 vs 70 | Matched H and T2DM individuals. 70% > 56 years or older; only 8% younger than 45 years | T2DM association with urinary IL-8 and evaluation of the effect of diet on urinary microbiota | NS | NS | Ruminococcus was positively associated with urinary IL-8 (results from Ling et al. 2017). Water weakened the positive relationship between Ruminococcus and IL-8 in urine while fiber, vitamin B3 and vitamin E enhanced the positive relationship. |
| Case-control studies with individuals that have kidney or renal disease | | | | | | | | |
| [30] | Portu-gal | 46 | 0 vs 14 | 57 ± 11 | All individuals with CKD and on peritoneal dialysis; the rest had other comorbidities | ↓ α–diversity (Shannon index) as CKD-PD worsened (independently of the cause) | ↓Atopobium ↓Dermabacter ↓Gardnerella | Apart from diabetes in CKD patients, significance was found for sex: Lactobacillus linked to females; Staphylococcus and Anaerococcus to males. |

(Continued)

**Table 1.** (Continued)

| Article | Country | Total adult | Healthy vs diabetic adults | Age: mean ± SD | Context | Indexes results in diabetic individuals | Main findings and/or differences in relative abundance of diabetic individuals in urine | Other significant details or relevant information |
|---------|---------|-------------|----------------------------|----------------|---------|------------------------------------------|----------------------------------------------------------------------------------------|--------------------------------------------------|
| [33] | United States | 77 | 23 vs 54 | Males: 77 ± 8 Females: 71 ± 8 | All individuals with CKD but no dialysis (70% of them with diabetes) | ↓ $\alpha$−diversity (Shannon index) as CKD worsened (independently of cause). Richness and $\alpha$−diversity were no different between diabetic and non-diabetic CKD individuals (Chao; Inverse Simpson and Shannon indexes). | NS | Richness and $\alpha$−diversity were no different by age or obesity status. β−diversity: Different urotypes. Urinary incontinence and ↑ eGFR was associated with ↑ $\alpha$−diversity. |

BMI = body mass index; CKD = chronic kidney disease; eGFR = estimated glomerular filtration rate; ESRD = end stage renal disease; FBG = fasting blood glucose; H = healthy controls;

HbA1c = glycosylated haemoglobin; IL–8 = interleukin 8; NS = Not stated; OTU = operational taxonomic unit; PD = peritoneal dialysis; SBP = systolic blood pressure; (TX)DM = (type X = 1 or

X = 2) Diabetes mellitus; SD = standard deviation; vs = versus;

* = studies that used data from participants/cohort of article 13 authored by Liu et al. 2017 [26];

ɛ = sequencing of microbiota extracellular vesicles;

¥ = primers for phylum Firmicutes; and genera *Bacteroides* and *Bifidobacterium*;

Ω = some or all individuals used antibiotic prophylaxis.

♀ = only females;

♂ = only males;

↓ = lower;

↑ = higher.

small sample sizes with less than 100 individuals, including five studies with less than 50 participants (Table 1). The only one that used shotgun sequencing had a small sample size (n = 29) (S2 Table in S1 File). Participants of these 18 studies were adults; individuals with diabetes ranged between 50 and 72 years old on average, and healthy controls were similar in age but slightly younger (if age was significantly different between groups, it is stated in the comments in Table 1). Selected studies had comprehensive exclusion criteria which are detailed in S2 Table in S1 File and they were assessed for risk of bias in S3 Table in S1 File. Moreover, as articles considered confounding factors, such as age, BMI or other relevant clinical factors; those that were statistically significant between groups are pointed up in the comments of Table 1.

## 3.2 Diversity and richness indexes

Among the 11 case-control studies, 9 reported richness and α-diversity analyses of which three of the studies observed a decrease in richness and α-diversity, one a decrease in richness, two a decrease in α-diversity, one observed an increase in α-diversity but this study assessed kidney stones patients with and without diabetes, and two studies did not observe differences in richness or α-diversity in the urobiome of the individuals with diabetes compared to those without diabetes (Table 1). Among the studies assessing only diabetic individuals, one study observed lower α-diversity in diabetic individuals with pyuria, one study observed lower richness in patients with diabetic peripheral neuropathy, one study did not observed differences in α-diversity when assessing urinary interleukin-8; and one study observed increased richness in diabetic individuals compared to diabetic individuals with hyperlipidaemia but decreased richness if compared to diabetic individuals with hypertension. Lastly, the two studies assessing CKD reported a decrease of α-diversity as CKD worsened independently of the primary cause of CDK (diabetes or other). Indexes calculated in each study are detailed in S4 Table in S1 File and the main ones are detailed in Table 1. Only seven studies reported β-diversity analyses with variable results (Table 1); 4 reported differences in β-diversity, of which 2 described different urotypes [33–36]; while 3 did not see differences in β-diversity [27, 37, 38].

## 3.3 Differences in microbial taxa

Significant results in microbial species for diabetic adults (statistical significance set to p< 0.05) of each article are summarised in Table 1 and all details can be found in S5 Table in S1 File. Comprising all studies, most of the bacterial species were identified at the genus level; few at species level and some at higher taxonomy levels. The relative abundance of 69 different bacteria was decreased, of 53 was increased in the urine of diabetic individuals while bacteria such as *Enterococcus*, *Stenotrophomonas*, *Klebsiella* or Actinobacteriota had increased or decreased relative abundance depending on studies (S5 and S6 Tables in S1 File and Table 1).

Overall, we could observe that the genus *Lactobacillus*, and at species level *Lactobacillus iners*, had increased relative abundance in diabetic individuals in 6 and 2 studies, respectively; and it was related to poor glycaemic control and with presence of interleukin-8 in urine [27, 28, 34–36, 39]. Three studies reported a decrease in relative abundance of the family *Bifidobacteriaceae* in urine of individuals with diabetes which includes the genera *Bifidobacterium* and *Gardnerella*; while one study reported the absence of the genus *Gardnerella* in individuals with diabetes and hyperlipidemia (note: *Gardnerella* is included now in the genus *Bifidobacterium*) [27, 28, 30, 40]. In addition, depletion of the family *Peptoniphilaceae* in urine of T2DM patients was reported in several selected studies. In fact, studies reported decreased relative abundance of this family being relevant genera *Peptoniphilus*, *Anaerococcus*, *Finegoldia*, *Murdochiella* and *Gallicola*, which were reported decreased abundant in four, three, two, one and one article, respectively [26, 27, 35, 36]. The genus *Parvimonas*, which belongs to this family as well, was

increased in relative abundance in diabetic individuals compared to non-diabetics only in the elderly cohort [39]; while its relative abundance was increased in the diabetic cohort compared to the diabetic plus hypertension cohort [28]. Lastly, three studies reported increased relative abundance of *Escherichia* (*Escherichia-Shigella*), a well-known genus causing urinary tract infections (UTI), which was related to poor glycaemic control and in two studies this genus was related to urinary symptoms or pyuria in urine (even though individuals did not have an active UTI) [32, 35, 41].

Taxonomic classification of bacteria identified in selected studies was updated according to Genome Taxonomy Database release 220 (https://gtdb.ecogenomic.org/) at higher taxonomical levels (e.g. phylum) but at lower levels was maintained to preserve clarity of the conclusions. For the most accurate taxonomic annotation refer to details in S6 Table in S1 File.

## 4. Discussion

This study has assessed the urinary microbiota related to diabetes and diabetes-related kidney diseases through a systematic review. This work has summarised statistically significant results reported by selected studies with a threshold of $p \leq 0.05$. The main findings of this study were: (1) a decreased richness and α-diversity in urinary microbiota in individuals with diabetes compared to healthy controls; (2) α-diversity was not different within diabetic individuals, except if they showed urinary symptoms (e.g. urgency to pee) although they did not have a (diagnosed) urinary infection; (3) α-diversity decreases with the evolution of kidney disease independently of the primary cause (e.g. kidney disease primarily caused by diabetes or hypertension or other pathology); (4) β-diversity urinary microbiota composition between healthy controls and diabetic individuals results were conflicting between studies and may depend on the stage of the disease and/or the grade of kidney damage; (5) patients with T2DM shared the depletion of *Peptoniphilus*, *Anaerococcus*, *Finegoldia* and *Gallicola* (all formerly part of the genus *Peptostreptococcus* and inside the *Peptoniphilaceae* family), or *Lachnospiraceae* that are butyrate-producing bacteria; (6) patients with diabetes shared the enrichment of pro-inflammatory microbes such as Pseudomonadota (formerly Proteobacteria) and of bacteria of order Lactobacillales such as *Lactobacillus* or *Enterococcus*; (7) cohort sizes were small (from a few samples to more than one hundred) and the only one study that was large (nearly seven hundred samples) had data restricted to extracellular microbial vesicles; (8) except in two studies, voided midstream urine samples were analysed and those are often referred as urogenital samples due to likely cross-contamination with skin or genitalia microbiota during collection; and, (9) although data showed similar trends, some bacterial phyla, families, and genera had controversial results, probably due to the lack of resolution of these studies using 16S sequencing.

One of the main findings of this systematic review is the depletion of the family *Peptoniphilaceae* in urine of T2DM patients as a trend. The most relevant genera of this family for the urobiome being *Peptoniphilus*, *Anaerococcus*, *Finegoldia*, *Murdochiella*, *Parvimonas* and *Gallicola*. These genera are gram-positive obligate anaerobic cocci that have been described as producers of short-chain fatty acids such as butyrate (*Peptoniphilus*, *Anaerococcus*, and *Gallicola*) and acetate (*Peptoniphilus*, *Parvimonnas*, *Gallicola* and *Finegoldia)* or lactate *(Murdochiella)* [44]. *Parvimonas* increased in relative abundance in diabetic individuals compared to non-diabetics only in the elderly cohort [39]; while its relative abundance was increased in the diabetic cohort compared to the diabetic plus hypertension cohort [28]. In the latter, if diabetes is accompanied by other factors such as hypertension, there is decreased abundance of this genus pointing to a depletion as the health status worsens. *Peptoniphilus* and *Anaerococcus* abundance may be associated to glycosylated haemoglobin HbA1c as diabetic individuals with high HbA1c had decreased relative abundance of these two genera in urine compared to

diabetic individuals with low HbA1c [35]. Generally, adults with diabetes and/or CKD show lower abundance of short-chain fatty acids producing bacteria in the gut compared to healthy controls [4, 45]. This systematic review reveals that diabetic individuals have similar trends in their urinary and urogenital microbiota with lower relative abundance of short-fatty acid butyrate-producing bacteria of the family *Peptoniphilaceae*. Butyrate is the main source of colonocyte metabolism, it is an enhancer of the integrity of the epithelium, and it decreases inflammation in the mucosa while promoting electrolyte absorption in the gut [46]. Moreover, dissociated butyric acid has an antibacterial effect [46] and butyryl-CoA regulates transcription and modifies proteins [46, 47]. We hypothesize that butyrate could also promote epithelium integrity and reduce inflammation in the urinary tract. Hence, depletion of butyrate-producing bacteria such as *Peptoniphilus*, *Anaerococcus*, and *Gallicola* in diabetic individuals could induce inflammation and contribute to the high onset of urinary infections in diabetic individuals in the urinary tract and induce inflammation in the renal blood vessels which could worsen kidney disease. This theory will need further studies to be tested, and it could help to understand if these molecules could be used to improve urinary functionality in association with diabetes and kidney disease.

Higher relative abundance of *Lactobacillus* associated with diabetes was a common trend of this systematic review. *Lactobacillus* have been described as a genus that is common in the urobiome of females and whose difference between females and males has been attributed to the hormone oestrogen, which plays a crucial role in glycogen production [11, 48]. Oestrogen fluctuations with age and menopausal status may affect the female urobiome [49], as the genus *Lactobacillus* metabolizes glycogen, and as a byproduct of it there is lactic acid which acidifies the environment modulating the growth of other bacteria [11]. The acidic environment has been assumed significant for the protective role that lactobacilli play in both the vaginal and urinary microbiomes of women [9, 48]. However, the genus *Lactobacillus* seems to be more abundant in the urine of diabetic individuals according to this systematic review. In line with these findings, history of diabetes was associated with increased odds of growing Lactobacilli when using expanded quantitative urine culture in catheterised urine of women [50]. *Lactobacillus iners*, which has been described as a transitional species that colonizes the vagina after an ecological disturbance [51], was found enriched in the urine of T2DM females [28, 39] and in the same cohort with higher levels of inflammatory interleukin-8 in urine [27]. Other species of this genus such as *Lactobacillus crispatus* and *Lactobacillus gasseri* have been associated with lack of lower urinary tract symptoms and lack of urgency urinary incontinence, respectively; and levels of *Lactobacillus* in the vagina are reported to be associated with menopausal status and with reduced risk of UTI [36]. Although lactobacilli have generally been considered beneficial and the normal microbiota of the urobiome and vagina, there is increasing evidence that some *Lactobacillus* spp. may be pathogenic or related to urinary symptoms [52, 53]. The association between the different species of *Lactobacillus* and their role in T2DM needs further investigation, considering that age, oestrogen, HbA1c and/or BMI could also be important factors affecting the abundance of this genus. In this context, HbA1c was positively correlated with *Lactobacillus*. Individuals with T2DM and high HbA1c had higher abundance of this genus compared to those with T2DM and low HbA1c levels [35, 36]. Another point to consider is that the genus *Lactobacillus* is naturally resistant to widely used antibiotics and T2DM individuals often have multiple infections that require the use of them [35]. Thus, we could hypothesize that antibiotic therapy may imbalance the urobiome allowing *Lactobacillus* to grow excessively which may induce inflammation.

Diabetic, prediabetic and obese individuals had lower abundance of the genus *Bifidobacterium* in urine; and the abundance of this genus was negatively correlated with age [39, 40]. In this line, T2DM patients with hyperlipidemia lacked *Bifidobacterium* (former *Gardnerella*)

[28]. This was also true for the former genus *Gardnerella*, that is now included in the genus *Bifidobacterium*, that have decreased abundance in CKD patients with diabetes [30]. *Bifidobacterium* (*Gardnerella*) was a common urotype in healthy females compared to diabetic females that had *Lactobacillus* or *Enterobacteriaceae* urotypes [36]. *Bifidobacteriaceae* had decreased relative abundance in diabetic females without interleukin-8 in urine in the study by Ling et al; but this group had also lower age and BMI compared to the group with interleukin-8 [27]. These results may imply that diabetes but also age, BMI or levels of interleukin-8 could influence the relative abundance of *Bifidobacterium* in urine [27].

The role of the genus *Prevotella* in the urinary microbiota is unclear. Although HbA1c was negatively correlated with *Prevotella* in T2DM patients [36] and Chen et al. reported a decrease of this genus in individuals with diabetes with high urinary symptoms [35], other studies have found that this genus was more prevalent in T2DM [42] or that urotypes and enrichment of *Prevotella* was associated with urinary incontinence patients [12, 54]. Therefore, similarly to findings with the genus *Lactobacillus*, we will likely need higher resolution studies, preferably shotgun metagenomics, to identify which species or strains of these genera may play different roles in the urobiome.

UTIs are common in diabetic patients and are associated with poor glycaemic control, which compromises kidney function and impairs quality of life [55]. That is why many selected articles also examined urinary symptoms, as hospitalization for UTIs caused by diabetes is more than twice as common as hospitalizations for UTIs caused by other factors. The use of sodium–glucose cotransporter 2 inhibitors as therapy for diabetes may contribute as these drugs reduce glucose reabsorption, thus lowering the level on blood, but causing glycosuria in urine which could increase the risk of UTIs in those patients [56]. Hyperglycaemia and diabetes have shown to reduce psoriasin and occludins levels in the urinary tract leading to be more prone to urinary tract infections [19]. Moreover, psoriasin is an antimicrobial peptide with a broad protective role against *Escherichia coli* and *Enterococcus* urinary infections, two of the most common pathogens causing UTI in diabetics [19]. In addition, low-grade inflammation with higher concentration of proinflammatory cytokines and the depletion of producers of short-chained fatty acids such as butyrate, which has beneficial effects on the host [57], may affect mucosa integrity, bacterial translocation and host–urine microbiota interactions in T2DM patients.

The results of this systematic review have also highlighted that urinary microbiota diversity seems lower in CKD or ESRD and it drops as the disease progresses independently of the primary cause [30, 33]. Diabetes progression to kidney damage reduces eGFR and as eGFR declines the excretion of uromodulin follows the same trend. Uromodulin, an urinary antimicrobial peptide whose function is to stick to bacteria forming larger particles which enhance their excretion by the kidney, is produced exclusively in the renal tubules and is affected by the decline of renal function [33] which may also influence bacterial growth, microbiota diversity and risk of UTI.

## 4.1 Strengths and limitations

To our knowledge, this review represents the first comprehensive analysis of the current understanding of the urobiome, specifically the urinary microbiota, in the context of diabetes. One notable strength of this review is the identification of emerging trends within the urobiome across multiple studies where the majority had low risk of bias assessment. For instance, this is the first article to highlight the significance of the *Peptoniphilaceae* family within the urobiome of diabetic individuals, which has frequently been reported as less prevalent or reduced in relative abundance in urine of diabetic individuals. Additionally, this work is strengthened by its systematic approach, adhering to PRISMA and PROSPERO guidelines, ensuring standardized and reproducible results.

However, we also recognize some limitations. First, we were unable to perform meta-analyses due to the considerable heterogeneity of the included studies and the small sample size in most of them. Second, the included studies have some methodological limitations such as the collection method of the urine, the small sample size of the studies, being the vast majority cross-sectional, and mostly they processed samples using 16S sequencing. Only two studies analysed catheterized urine, reflecting the microbiota of the bladder or renal pelvis, while most of the studies relied in voided midstream collection techniques that may also capture microbiota from the genitalia or skin (Linda Brubaker et al., 2021). Since most studies employed 16S sequencing, the resolution of the results is limited and there may be discrepancies depending on the region sequenced (most of them used the V3-V4 region or only the V4 region of the 16S rDNA gene). This limitation means that the same genera identified across different studies could refer to distinct bacterial species, potentially explaining some discrepancies in the findings. Moreover, microbial taxa such as fungi or small prokaryotic taxa that may also contribute to the urobiome community are often not captured in these studies, as they must sequence the eukaryotic 18S rDNA gene. Another limitation is the evolving taxonomical classification of bacteria. Taxonomic reclassification may have occurred since the original articles were published. Where possible, we have updated higher-level classifications (e.g., from Firmicutes to Bacillota) while maintaining the original nomenclature at lower levels to preserve the clarity of the conclusions (e.g., *Gardnerella* instead of *Bifidobacterium*). For the most accurate taxonomic annotation, we refer readers to the Genome Taxonomy Database release 220 (https://gtdb.ecogenomic.org/), with detailed information provided in S6 Table in S1 File. Therefore, the generality of these findings may be treated as trends, with caution, and these hypotheses should be tested in larger size studies and analysed using high resolution shotgun sequencing.

## 5. Future perspectives

There are significant knowledge gaps regarding the role and impact of the urobiome in both health and disease. This emerging field has attracted increasing attention in recent years, but the influence of the urobiome on conditions such as diabetes and diabetic kidney disease remains largely unexplored. While this systematic review has identified common trends across studies—such as the depletion of *Peptoniphilaceae* and the increased abundance of *Lactobacillus* in the urine of diabetic adults—the evidence remains insufficient due to the small sample sizes of most studies and the limited resolution provided by 16S sequencing.

To advance urobiome research, large-scale population-based studies are urgently needed, along with the application of higher-resolution techniques like shotgun metagenomics to more precisely characterize the urinary microbiome. Longitudinal studies will also help to understand the changes in the urinary microbiome in diabetes and diabetic kidney disease progression, including the role of the urobiome in early stages of diabetes development. These approaches could identify generalized results including specific microbial species that play critical roles in maintaining a healthy urobiome, as well as those involved in the onset and progression of diseases such as diabetes and conditions affecting kidney function. In the future, the applicability of this knowledge may help us to investigate the influence of diet on the urobiome in diabetes and the potential for probiotics, postbiotics or other interventions to modulate the urobiome in relation to diabetes and diabetic kidney disease.

## Supporting information

**S1 File.**
(DOCX)

**S1 Checklist. PRISMA 2020 checklist.**
(DOCX)

**S1 Table. Original records retrieved from Pubmed, SCOPUS, MEDLINE-Ovid, EMBASE, Web of Science and Cochrane Central Register of Controlled Trials databases.**
(XLSX)

## Author Contributions

**Conceptualization:** Tiscar Graells.

**Data curation:** Tiscar Graells, Yi-Ting Lin.

**Formal analysis:** Tiscar Graells, Yi-Ting Lin.

**Funding acquisition:** Shafqat Ahmad, Tove Fall, Johan Ärnlöv.

**Investigation:** Tiscar Graells.

**Methodology:** Tiscar Graells.

**Supervision:** Tiscar Graells, Johan Ärnlöv.

**Validation:** Tiscar Graells, Yi-Ting Lin, Johan Ärnlöv.

**Writing – original draft:** Tiscar Graells.

**Writing – review & editing:** Tiscar Graells, Yi-Ting Lin, Shafqat Ahmad, Tove Fall, Johan Ärnlöv.

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
