## [Decision Letter · Decision Letter 0]

19 Nov 2024

PONE-D-24-45699The urinary microbiome in association with diabetes and diabetic kidney disease: A systematic reviewPLOS ONE

Dear Dr. Graells,

Thank you for submitting your manuscript to PLOS ONE. After careful consideration, we feel that it has merit but does not fully meet PLOS ONE’s publication criteria as it currently stands. Therefore, we invite you to submit a revised version of the manuscript that addresses the points raised during the review process.

We look forward to receiving your revised manuscript.

Kind regards,

Zongxin Ling

Academic Editor

PLOS ONE

Journal Requirements:

1. When submitting your revision, we need you to address these additional requirements. Please ensure that your manuscript meets PLOS ONE's style requirements, including those for file naming. The PLOS ONE style templates can be found athttps://journals.plos.org/plosone/s/file?id=wjVg/PLOSOne_formatting_sample_main_body.pdf andhttps://journals.plos.org/plosone/s/file?id=ba62/PLOSOne_formatting_sample_title_authors_affiliations.pdf 2. Thank you for stating the following financial disclosure: [We acknowledge funding from the Swedish research council (Vetenskapsrådet grant number 2019-01015, 2019-01471 and 2020-0243), from the Swedish heart lung foundation (Hjärt-Lungfonden grant number 2021-0357 and 2023-0687), and from Center of clinical research (CKF) in Region Dalarna, Falun, Sweden. Dr Ahmad was supported by research grants from FORMAS –Early Career Grant (no. 2020-00989), Swedish Research Council-Early Career Grant (no. 2022-01460), EFSD/Novo Nordisk and EpiHealth]. Please state what role the funders took in the study. If the funders had no role, please state: "The funders had no role in study design, data collection and analysis, decision to publish, or preparation of the manuscript." If this statement is not correct you must amend it as needed. Please include this amended Role of Funder statement in your cover letter; we will change the online submission form on your behalf. 3. We note that there is identifying data in the Supporting Information file <Supp_information_SysRev_DM_Urobiome.docx>. Due to the inclusion of these potentially identifying data, we have removed this file from your file inventory. Prior to sharing human research participant data, authors should consult with an ethics committee to ensure data are shared in accordance with participant consent and all applicable local laws. Data sharing should never compromise participant privacy. It is therefore not appropriate to publicly share personally identifiable data on human research participants. The following are examples of data that should not be shared: -Name, initials, physical address-Ages more specific than whole numbers-Internet protocol (IP) address-Specific dates (birth dates, death dates, examination dates, etc.)-Contact information such as phone number or email address-Location data-ID numbers that seem specific (long numbers, include initials, titled “Hospital ID”) rather than random (small numbers in numerical order) Data that are not directly identifying may also be inappropriate to share, as in combination they can become identifying. For example, data collected from a small group of participants, vulnerable populations, or private groups should not be shared if they involve indirect identifiers (such as sex, ethnicity, location, etc.) that may risk the identification of study participants. Additional guidance on preparing raw data for publication can be found in our Data Policy (https://journals.plos.org/plosone/s/data-availability#loc-human-research-participant-data-and-other-sensitive-data) and in the following article: http://www.bmj.com/content/340/bmj.c181.long. Please remove or anonymize all personal information (name, initials), ensure that the data shared are in accordance with participant consent, and re-upload a fully anonymized data set. Please note that spreadsheet columns with personal information must be removed and not hidden as all hidden columns will appear in the published file. 4. Please include captions for your Supporting Information files at the end of your manuscript, and update any in-text citations to match accordingly. Please see our Supporting Information guidelines for more information: http://journals.plos.org/plosone/s/supporting-information. 5. As required by our policy on Data Availability, please ensure your manuscript or supplementary information includes the following:  A numbered table of all studies identified in the literature search, including those that were excluded from the analyses.   For every excluded study, the table should list the reason(s) for exclusion.   If any of the included studies are unpublished, include a link (URL) to the primary source or detailed information about how the content can be accessed.  A table of all data extracted from the primary research sources for the systematic review and/or meta-analysis. The table must include the following information for each study:  Name of data extractors and date of data extraction  Confirmation that the study was eligible to be included in the review.   All data extracted from each study for the reported systematic review and/or meta-analysis that would be needed to replicate your analyses.  If data or supporting information were obtained from another source (e.g. correspondence with the author of the original research article), please provide the source of data and dates on which the data/information were obtained by your research group.  If applicable for your analysis, a table showing the completed risk of bias and quality/certainty assessments for each study or outcome.  Please ensure this is provided for each domain or parameter assessed. For example, if you used the Cochrane risk-of-bias tool for randomized trials, provide answers to each of the signalling questions for each study. If you used GRADE to assess certainty of evidence, provide judgements about each of the quality of evidence factor. This should be provided for each outcome.   An explanation of how missing data were handled.  This information can be included in the main text, supplementary information, or relevant data repository. Please note that providing these underlying data is a requirement for publication in this journal, and if these data are not provided your manuscript might be rejected.  

Reviewers' comments:

Reviewer's Responses to Questions

**Comments to the Author**

1. Is the manuscript technically sound, and do the data support the conclusions?

Reviewer #1: Yes

Reviewer #2: Partly

Reviewer #3: Yes

2. Has the statistical analysis been performed appropriately and rigorously? 

Reviewer #1: Yes

Reviewer #2: N/A

Reviewer #3: Yes

3. Have the authors made all data underlying the findings in their manuscript fully available?

Reviewer #1: Yes

Reviewer #2: Yes

Reviewer #3: Yes

4. Is the manuscript presented in an intelligible fashion and written in standard English?

Reviewer #1: Yes

Reviewer #2: Yes

Reviewer #3: Yes

5. Review Comments to the Author

Reviewer #1: The authors reviewed urinary microbiome in patients with diabetic kidney disease. This review is valuable. However, there are still some issues.

1. Line 91. As a systematic review, SCOPUS and PUBMED databases are not enough. Databases such as PubMed/Medline, Web of Science, EMBASE, and the Cochrane Central Register of Controlled Trials should be included.

2. The abundance of urine microbiome is relatively low, so different detection methods may bring large errors. There is a big gap between the technology of 20 years ago and the technology of today, which may produce a large bias in the analysis. It is suggested that the authors change the search range to nearly 10 years. For example, from 2014 to 2024.

3. There is only one figure and one table in the manuscript. Why did the authors include so much data in the supplementary material? Some data should be moved from the supplementary material.

Reviewer #2: This is a well-structured and informative review of the current knowledge on the urobiome and its association with diabetes mellitus and diabetic kidney disease. The review effectively highlights the emerging field of urobiome research and its potential implications for understanding and managing kidney disease.

Here are some areas for potential improvement:

While the review provides a good overview of the trends in the literature, a deeper analysis of the specific microbial taxa and functional pathways associated with diabetes and kidney disease could be beneficial.

The review could discuss in more detail the methodological limitations of the included studies, such as potential biases in sample collection and processing, and the impact of these limitations on the overall findings.

The review could conclude with a discussion of future research directions, including the need for larger, well-designed studies to confirm the findings and explore the mechanisms underlying the associations between the urobiome and diabetes.

While you summarize results on richness, alpha/beta diversity, and specific bacterial taxa, consider offering a more interpretive analysis. Explore potential connections between specific microbial changes and diabetes or DKD.

Discuss the limitations of the included studies, such as small sample sizes, cross-sectional design, and heterogeneity in sample collection methods. Consider how these limitations might affect the generalizability of the findings.

Conclude by outlining potential future research directions. You could expand on specific areas of future research. This could include

• investigating the influence of specific dietary patterns on the urobiome in diabetes.

• exploring the potential for manipulating the urobiome through probiotics or other interventions.

• investigating the role of the urobiome in the early stages of diabetes development.

You could briefly mention the potential implications of these findings for understanding and managing diabetes and DKD.

Briefly discuss potential mechanisms by which the observed changes in the urobiome might influence diabetes and DKD. For example, how could the decrease in butyrate-producing bacteria contribute to disease?

Please provide a full search strategy for all databases.

Reviewer #3: Graells et al. have performed a systematic review on the association of urinary microbiome with diabetes and diabetic kidney diseases. The study findings are interesting and the manuscript is well-written. These are my comments:

- Abstract should have a structure of background, methods, results, and conclusion.

- The introduction section is rather long in its current format. The authors should focus on the main ideas related to the topic and try to emphasize the gaps in knowledge.

- The results section could be organized by adding subheadings.

- A supplementary table containing the search strategy and the exact words searched in each database could be helpful for the reproducibility of the study.

6. PLOS authors have the option to publish the peer review history of their article (what does this mean?). If published, this will include your full peer review and any attached files.

Reviewer #1: No

Reviewer #2: No

Reviewer #3: No

---

## [Author Response · Author response to Decision Letter 0]

19 Dec 2024

Reviewers' comments:

Reviewer's Responses to Questions

Comments to the Author

1. Is the manuscript technically sound, and do the data support the conclusions?

Reviewer #1: Yes

Reviewer #2: Partly

Reviewer #3: Yes

2. Has the statistical analysis been performed appropriately and rigorously?

Reviewer #1: Yes

Reviewer #2: N/A

Reviewer #3: Yes

3. Have the authors made all data underlying the findings in their manuscript fully available?

Reviewer #1: Yes

Reviewer #2: Yes

Reviewer #3: Yes

4. Is the manuscript presented in an intelligible fashion and written in standard English?

Reviewer #1: Yes

Reviewer #2: Yes

Reviewer #3: Yes

5. Review Comments to the Author

Reviewer #1: The authors reviewed urinary microbiome in patients with diabetic kidney disease. This review is valuable. However, there are still some issues.

1. Line 91. As a systematic review, SCOPUS and PUBMED databases are not enough. Databases such as PubMed/Medline, Web of Science, EMBASE, and the Cochrane Central Register of Controlled Trials should be included.

Authors’ response: Thank you for pointing out that fact. We have updated the Methods Search Strategy section to include these new database searches and to update the previous searches. We have now included until the end of November a search in MEDLINE Ovid, Web of Science and EMBASE where we found 38,86 and 87 records, respectively. Duplicates were 37, 45 and 63 in MEDLINE Ovid, Web of Science and EMBASE, respectively. One, 17 and 21 were articles types in MEDLINE Ovid, Web of Science and EMBASE, respectively, not included in our analyses such as reviews, conference abstracts. Twenty-one and three articles were excluded after screening title and abstract in Web of Science and EMBASE, respectively. Hence, in the end only three articles were eligible for full screening from Web of Science. In the end, none of them were included in the analysis after full screening as they did not fulfil the inclusion criteria. 

We have also searched the Cochrane Central Register of Controlled Trials where there were no matching for Reviews, Protocols, Trials, Editorials, Special collections or Clinical answers following a TITLE-ABS-KEY search [[ 0 Cochrane Reviews matching "urobiome" OR "urinary microbiome" OR "urinary microbiota" OR "urinary tract microb*" OR "urine microb*" OR "urogenital microbiome" OR "urogenital microbiota" in Title Abstract Keyword AND "kidney disease*" OR "chronic kidney disease" OR "diabetes" OR "albuminuria" OR "diabetic nephropathy" OR "end stage renal disease*" OR "renal disease*" in Title Abstract Keyword - (Word variations have been searched) ]]. All this is specified in the Methods search strategy.

2. The abundance of urine microbiome is relatively low, so different detection methods may bring large errors. There is a big gap between the technology of 20 years ago and the technology of today, which may produce a large bias in the analysis. It is suggested that the authors change the search range to nearly 10 years. For example, from 2014 to 2024.

Authors’ response: Thank you for this pointing this fact. It is completely true that technology has changed a lot in 20 years. And, we could have searched only one decade, but we did not want to miss anything that could have been published. However, the first article that fulfilled the inclusion criteria was published only 7 years ago, in 2017. Hence, although technology has changed in 20 years, we believe that this is not affecting our analysis as articles selected following the inclusion criteria are from 2017 onwards.

3. There is only one figure and one table in the manuscript. Why did the authors include so much data in the supplementary material? Some data should be moved from the supplementary material.

Authors’ response: Thank you for noticing that we have conducted a thorough analysis of each of the articles. We have presented all data we have checked for each of the articles in the supplementary information. We believe that the summarized and key information is the one presented in Table 1. That is why, Table 1 is so extensive, and we decided still to make available all the data collected in the supplementary information for interested researchers. We believe that the article is easier and more attractive to read by summarizing the most important information in a single table in the main manuscript. However, we are willing to reorganize if the Editor agrees that this would be an improvement of the manuscript.

Reviewer #2: This is a well-structured and informative review of the current knowledge on the urobiome and its association with diabetes mellitus and diabetic kidney disease. The review effectively highlights the emerging field of urobiome research and its potential implications for understanding and managing kidney disease.

Here are some areas for potential improvement:

While the review provides a good overview of the trends in the literature, a deeper analysis of the specific microbial taxa and functional pathways associated with diabetes and kidney disease could be beneficial.

Authors’ response: Thank you for this comment. We have expanded the discussion and references with the following. “Generally, adults with diabetes and/or chronic kidney disease show lower abundance of short-chain fatty acids producing bacteria in the gut compared to healthy controls [4,45]. This systematic review reveals that diabetic individuals have similar trends but also differences in their urinary and urogenital microbiota with lower relative abundance of short-fatty acid butyrate-producing bacteria of the family Peptoniphilaceae. Butyrate is the main source of colonocyte metabolism, it is an enhancer of the integrity of the epithelium, and it decreases inflammation in the mucosa while promoting electrolyte absorption in the gut [46]. Moreover, dissociated butyric acid has an antibacterial effect [46] and butyryl-CoA regulates transcription and modifies proteins [46,47]. We hypothesize that butyrate could also promote epithelium integrity and reduce inflammation in the urinary tract. Hence, depletion of butyrate-producing bacteria such as Peptoniphilus, Anaerococcus, and Gallicola in diabetic individuals could induce inflammation and contribute to the high onset of urinary infections in diabetic individuals in the urinary tract and induce inflammation in the renal blood vessels which could worsen kidney disease. This theory will need further studies to be confirmed but it could help to understand if the use of butyrate as a postbiotic could improve urinary functionality in association with diabetes and kidney disease.”

The review could discuss in more detail the methodological limitations of the included studies, such as potential biases in sample collection and processing, and the impact of these limitations on the overall findings.

Authors’ response: Thank you for this comment. We have expanded the limitations section with lines 335-343.

The review could conclude with a discussion of future research directions, including the need for larger, well-designed studies to confirm the findings and explore the mechanisms underlying the associations between the urobiome and diabetes.

Authors’ response: Thank you for this comment. We have expanded the future perspectives section with lines 358-367.

While you summarize results on richness, alpha/beta diversity, and specific bacterial taxa, consider offering a more interpretive analysis. Explore potential connections between specific microbial changes and diabetes or DKD.

Authors’ response: Thank you for this comment. We have expanded the discussion and references with the following. “Generally, adults with diabetes and/or chronic kidney disease show lower abundance of short-chain fatty acids producing bacteria in the gut compared to healthy controls [4,45]. This systematic review reveals that diabetic individuals have similar trends but also differences in their urinary and urogenital microbiota with lower relative abundance of short-fatty acid butyrate-producing bacteria of the family Peptoniphilaceae. Butyrate is the main source of colonocyte metabolism, it is an enhancer of the integrity of the epithelium, and it decreases inflammation in the mucosa while promoting electrolyte absorption in the gut [46]. Moreover, dissociated butyric acid has an antibacterial effect [46] and butyryl-CoA regulates transcription and modifies proteins [46,47]. We hypothesize that butyrate could also promote epithelium integrity and reduce inflammation in the urinary tract. Hence, depletion of butyrate-producing bacteria such as Peptoniphilus, Anaerococcus, and Gallicola in diabetic individuals could induce inflammation and contribute to the high onset of urinary infections in diabetic individuals in the urinary tract and induce inflammation in the renal blood vessels which could worsen kidney disease. This theory will need further studies to be confirmed but it could help to understand if the use of butyrate as a postbiotic could improve urinary functionality in association with diabetes and kidney disease.”

Discuss the limitations of the included studies, such as small sample sizes, cross-sectional design, and heterogeneity in sample collection methods. Consider how these limitations might affect the generalizability of the findings.

Authors’ response: Thank you for this comment. We have expanded the limitations section with lines 335-343.

Conclude by outlining potential future research directions. You could expand on specific areas of future research. This could include

• investigating the influence of specific dietary patterns on the urobiome in diabetes.

• exploring the potential for manipulating the urobiome through probiotics or other interventions.

• investigating the role of the urobiome in the early stages of diabetes development.

Authors’ response: Thank you for this comment. We have expanded the future perspectives section with lines 358-367.

You could briefly mention the potential implications of these findings for understanding and managing diabetes and DKD.

Briefly discuss potential mechanisms by which the observed changes in the urobiome might influence diabetes and DKD. For example, how could the decrease in butyrate-producing bacteria contribute to disease? 

Authors’ response: Thank you for this comment. We have expanded the discussion and references with the following. “Generally, adults with diabetes and/or chronic kidney disease show lower abundance of short-chain fatty acids producing bacteria in the gut compared to healthy controls [4,45]. This systematic review reveals that diabetic individuals have similar trends but also differences in their urinary and urogenital microbiota with lower relative abundance of short-fatty acid butyrate-producing bacteria of the family Peptoniphilaceae. Butyrate is the main source of colonocyte metabolism, it is an enhancer of the integrity of the epithelium, and it decreases inflammation in the mucosa while promoting electrolyte absorption in the gut [46]. Moreover, dissociated butyric acid has an antibacterial effect [46] and butyryl-CoA regulates transcription and modifies proteins [46,47]. We hypothesize that butyrate could also promote epithelium integrity and reduce inflammation in the urinary tract. Hence, depletion of butyrate-producing bacteria such as Peptoniphilus, Anaerococcus, and Gallicola in diabetic individuals could induce inflammation and contribute to the high onset of urinary infections in diabetic individuals in the urinary tract and induce inflammation in the renal blood vessels which could worsen kidney disease. This theory will need further studies to be confirmed but it could help to understand if the use of butyrate as a postbiotic could improve urinary functionality in association with diabetes and kidney disease.”

Authors’ response: Thank you for this comment.

Please provide a full search strategy for all databases.

Authors’ response: Thank you for this comment. We have updated the Methods Search Strategy section to include new databases in this analysis. We have updated the information provided and we have made available our search strategy in the following lines: “A comprehensive literature search on different databases was conducted from 1st January 2000 until 28th November 2024. The search included the following terms [("urobiome" OR "urinary microbiome" OR "urinary microbiota" OR "urinary tract microb*" OR "urine microb*" OR "urogenital microbiome" OR "urogenital microbiota") AND ("kidney disease*" OR "chronic kidney disease" OR "diabetes" OR "albuminuria" OR "diabetic nephropathy" OR "end stage renal disease*" OR "renal disease*")]. We used the ‘TITLE-ABS-KEY’ search in SCOPUS and Cochrane Central Register of Controlled Trials databases, the ‘Keyword’ in advanced search in MEDLINE-Ovid and EMBASE databases, while ‘ALL FIELDS’ were used in Web of Science and PUBMED databases.”.

Reviewer #3: Graells et al. have performed a systematic review on the association of urinary microbiome with diabetes and diabetic kidney diseases. The study findings are interesting and the manuscript is well-written. These are my comments:

- Abstract should have a structure of background, methods, results, and conclusion.

Authors’ response: Thank you for this comment. Now, we have added these sections in the abstract. 

- The introduction section is rather long in its current format. The authors should focus on the main ideas related to the topic and try to emphasize the gaps in knowledge.

Authors’ response: In the revised version of the manuscript, we have shortened the introduction part. 

- The results section could be organized by adding subheadings.

Authors’ response: Thank you for this comment. Now, we have added these sections (3.1 Selected studies; 3.2 Diversity and richness indexes; and 3.3 Differences in microbial microbial taxa). 

- A supplementary table containing the search strategy and the exact words searched in each database could be helpful for the reproducibility of the study.

Authors’ response: Thank you for this comment. We have updated the Methods Search Strategy section to include new databases in this analysis. We have updated the information provided including a supplementary table with the details of the search startegy and we have made available our search strategy in the following lines: “A comprehensive literature search on different databases was conducted from 1st January 2000 until 28th November 2024. The search included the following terms [("urobiome" OR "urinary microbiome" OR "urinary microbiota" OR "urinary tract microb*" OR "urine microb*" OR "urogenital microbiome" OR "urogenital microbiota") AND ("kidney disease*" OR "chronic kidney disease" OR "diabetes" OR "albuminuria" OR "diabetic nephropathy" OR "end stage renal disease*" OR "renal disease*")]. We used the ‘TITLE-A

---

## [Decision Letter · Decision Letter 1]

8 Jan 2025

The urinary microbiome in association with diabetes and diabetic kidney disease: A systematic review

PONE-D-24-45699R1

Dear Dr. Graells,

We’re pleased to inform you that your manuscript has been judged scientifically suitable for publication and will be formally accepted for publication once it meets all outstanding technical requirements.

Kind regards,

Zongxin Ling

Academic Editor

PLOS ONE

Additional Editor Comments (optional):

Reviewers' comments:

Reviewer's Responses to Questions

**Comments to the Author**

1. If the authors have adequately addressed your comments raised in a previous round of review and you feel that this manuscript is now acceptable for publication, you may indicate that here to bypass the “Comments to the Author” section, enter your conflict of interest statement in the “Confidential to Editor” section, and submit your "Accept" recommendation.

Reviewer #1: All comments have been addressed

Reviewer #2: All comments have been addressed

Reviewer #3: All comments have been addressed

2. Is the manuscript technically sound, and do the data support the conclusions?

Reviewer #1: Yes

Reviewer #2: Yes

Reviewer #3: (No Response)

3. Has the statistical analysis been performed appropriately and rigorously? 

Reviewer #1: Yes

Reviewer #2: Yes

Reviewer #3: (No Response)

4. Have the authors made all data underlying the findings in their manuscript fully available?

Reviewer #1: Yes

Reviewer #2: Yes

Reviewer #3: (No Response)

5. Is the manuscript presented in an intelligible fashion and written in standard English?

Reviewer #1: Yes

Reviewer #2: Yes

Reviewer #3: (No Response)

6. Review Comments to the Author

Reviewer #1: Thank you very much. You guys solved all my problems. I think the present article is worth publishing.

Reviewer #2: the aim of this systematic review is to gather available data and synthetise the main

80 findings about the urobiome, particularly, the urinary microbiota in association with diabetes and

81 kidney conditions derived from diabetes such as CKD or ESRD. All of my comments have been responded satisfactorily.

Reviewer #3: (No Response)

7. PLOS authors have the option to publish the peer review history of their article (what does this mean?). If published, this will include your full peer review and any attached files.

Reviewer #1: No

Reviewer #2: No

Reviewer #3: No

---

## [Editor Report · Acceptance letter]

22 Jan 2025

PONE-D-24-45699R1 

PLOS ONE

Dear Dr. Graells, 

I'm pleased to inform you that your manuscript has been deemed suitable for publication in PLOS ONE. Congratulations! Your manuscript is now being handed over to our production team.

Kind regards, 

on behalf of

Dr. Zongxin Ling 

Academic Editor

PLOS ONE